# Enhancing the Anti-Migration Performance and Mechanical Properties of EPDM Insulation through Functionalized GO

**DOI:** 10.3390/polym15071731

**Published:** 2023-03-30

**Authors:** Zhehong Lu, Ziqiang Zhu, Yulong Zhang, Chenyang Wang, Haoran Bai, Guangpu Zhang, Yubing Hu, Wei Jiang

**Affiliations:** 1National Special Superfine Powder Engineering Research Center of China, Nanjing University of Science and Technology, Nanjing 210014, China; 2China North Industry Advanced Technology Generalization Institute, Beijing 100089, China; 3College of Materials Science and Engineering, Shenyang University of Technology, Shenyang 110870, China

**Keywords:** mechanical properties, graphene oxide, insulation, anti-migration

## Abstract

The excessive migration of small molecular plasticizers in solid propellants may lead to debonding and changes in combustion characteristics, affecting the safety of solid rocket motors. Herein, two functionalized graphene oxides (GO) were used to enhance the anti-migration performance of EPDM insulation. GO, 3-Aminopropyltriethoxysilane-modified GO (AGO) and octadecylamine-modified GO (HGO) were filled into EPDM to fabricate EPDM insulation. The anti-migration properties and migration kinetics of EPDM insulations were studied using immersion tests. Moreover, the mechanical properties, including the tensile properties, crosslink density, hardness, and aging resistance of different EPDM insulations, were also explored. Compared with GO, AGO, and HGO obviously enhanced the anti-migration and mechanical properties of the EPDM insulations. This study shows that the anti-migration performance of EPDM insulation can be enhanced by functionalized GO.

## 1. Introduction

Today, the solid rocket motor (SRM) is widely used in the aerospace field. The insulation is a thermal protection material located between the inner surface of the shell and the propellant in the SRM [1]. Its main function is to remove most of the heat through its continuous decomposition and ablation to slow down the propagation speed of the high temperature of the gas to the shell, avoid the shell reaching the temperature that endangers its structural integrity, and ensure the normal and safe operation of the SRM [2,3]. Due to the specific location and special functions of the internal insulation layer, there are strict requirements for the internal insulation materials, not only in terms of thermal properties and ablation resistance but also in terms of mechanical properties, compatibility, process performance, and anti-aging properties [4]. Ethylene propylene diene monomer (EPDM) is widely used as the combustion chamber of SRM owing to its outstanding age-resistant, ablate-resistant, thermal shock-resistant, and low-density properties [5,6,7].

During the storage of SRM, plasticizers such as nitroglycerin (NG) and dibutyl phthalate (DBP) in the grain will migrate into the insulation, changing the composition of the contact surface between the insulation and propellant, reducing energy and burning rate, and changing bond strength [8,9]. A more serious problem is to reduce the flame-retardant coating insulation effect, thus destroying the interior ballistic performance of the rocket engine. Accordingly, the improvement of anti-migration properties is urgently needed. Numerous efforts have been put into hindering the migration of plasticizers [10,11,12]. Some scholars [10] tried to reduce the diffusion coefficient by adding a barrier layer. The barrier layer, also known as the transition layer and the protective layer, is located between the propellant and the insulation. It has the function of preventing the migration of nitrate plasticizer and improving the bonding performance of the insulation. It is an important technology for the anti-migration of the propellant insulation. Hadi Rezaei-Vahidian et al. [11] used an inhibitor to prevent plasticizer migration from the polyurethane matrix to the EPDM-based substrate. The results showed that using β-cyclodextrin as an inhibitor agent in the PU binder could prevent the migration of plasticizer to the EPDM substrate. Zhang Bowen et al. [12] used toluene diisocyanat, hydroxyl-terminated polyester diol, and glycerin to fabricate a transition layer between the propellant and the insulation, which greatly inhibited the migration of NENA in the propellant. Wu Wei et al. [13] bonded an isolation coating between the solid propellant and the insulation to prevent the migration of nitroglycerin, which consisted of 3-aminopropyltriethoxysilane, phenolic resin, and polyvinyl butyral. The NG content in PU inhibitor was reduced from 4.5% to 0.48%, and the thermal analysis showed that the isolation coating significantly reduced the tendency toward NG migration. However, the process of this method is more complicated.

Another effective measure is incorporating inorganic fillers in the insulation, which can provide a mechanical barrier against NG migration. Meanwhile, it is also advantageous to enhance the mechanical properties of the insulation. The inorganic fillers can increase the density of the insulation, block the cross-linked mesh, and reduce the free volume of the material, which has a good effect on the inhibition of NG migration [14,15]. Tohid Farajpour et al. [16] fabricated a liner containing carboxylated carbon nanotube (CNT), which they utilized to reduce the migration of dioctylphthalate (DOP). They concluded that CNT prevents the migration of DOP by electrostatic interaction. Graphene is a two-dimensional residual nanomaterial composed of carbon atoms in sp^2^ hybrid orbitals with a hexagonal honeycomb lattice. Its internal benzene ring has a very high electron density, so that atoms and molecules cannot pass through. As a result, graphene has a high resistance to small molecule diffusion [17].

Graphene has garnered significant interest from researchers as a potential solution to the issue of molecular migration in recent years. A team from Beijing University of Chemical Technology Zhang Liqun [18] demonstrated that graphite-modified butyl rubber has excellent barrier properties, and its resistance to mustard gas diffusion tree decreased by 46% compared with unmodified butyl rubber. Al-Jabareen et al. [19] prepared PET/graphene composites to improve oxygen barrier properties. Due to the barrier properties of graphene and its effect on the crystallinity of the composites, the maximum oxygen transmittance of the composites was reduced by more than 99%. Lu et al. [20] added GO to the liners to increase the migration resistance of dioctyl sebacate, the enhanced anti-migration ability is due to the physical barrier of GO, which produces quality tortuous paths for diffusing small molecules [21,22].

Combined with the above research, it can be seen that researchers have carried out a large number of anti-migration properties of graphene theoretical research and experimental exploration and made outstanding progress in various fields, including food preservation, sealing, anti-corrosion, safety, etc. [23]. The migration of small molecules of plasticizer and nitroglycerin to the insulation layer has always been a pressing issue that needed to be addressed [24,25,26] The current research will be a good basis for improving the anti-migration characteristics of propellant insulation [27,28].

In this work, GO was added into EPDM to prepare the composite insulation. To further enhance the anti-migration performance and mechanical properties of the insulation, GO was functionalized by 3-aminopropyltriethoxysilane (APTES) and octadecylamine (ODA). Four different insulations were prepared as a comparison, namely EPDM, GO/EPDM, AGO/EPDM, and HGO/EPDM, respectively. The migration of mixed plasticizers to the different insulations were assessed via the immersion method. In addition, the mechanical properties of the insulations were also evaluated.

## 2. Experimental

### 2.1. Materials

EPDM (5565, 5.5 wt.% 5-ethylidene-2-norbornene (ENB)) was supplied by Dow Chemical (Midland, MI, USA). Flake graphite was purchased from obtained from Qingdao Tianhe Graphite Co., Ltd. (Qingdao, China). Potassium permanganate (KMnO_4_, AR), cyclohexane, concentrated sulfuric acid (H_2_SO_4_, 95–98%), deionized water (DI), hydrochloric acid (37%), triacetin, hydrogen peroxide (H_2_O_2_), and absolute ethanol were provided by Shanghai Macklin Biochemical Co., Ltd. (Shanghai, China) and used as received. APTES, ODA, DBP, and ethyl nitrate were analytically pure and supplied by Sinopharm Chemical Reagent Co., Ltd. (Shanghai, China) Sulfur, zinc oxide, and stearic acid were commercial products.

### 2.2. Preparation of AGO and HGO

GO was prepared by a modified Hummers method [29,30]. The synthesis route of AGO and HGO are shown in Figure 1 [29,30]. The detailed synthesis method is shown in the Appendix A.

### 2.3. Fabrication of EPDM Insulation Composites

In this step, EPDM insulation was prepared using the solution mixing method, which improved the filler’s dispersion. Firstly, EPDM was dissolved in cyclohexane to create an EPDM emulsion and then GO (3 phr), AGO (3 phr), and HGO (3 phr) were introduced and thoroughly mixed by an ultrasonic system for 30 min. Next, sulfur, zinc oxide, and other additives were introduced into the EPDM/fillers suspension. After the solvent was completely volatilized, the dried mixture was mixed in a two-roll mixer. Finally, the well dispersed EPDM insulation was obtained by vulcanization at 15 MPa and 160 °C for 30 min. EPDM insulation composites were recorded as EPDM, GO/EPDM, AGO/EPDM, and HGO/EPDM, respectively.

### 2.4. Characterization

XRD was performed on the D8 Advance (Bruker, Bremen, German), the measurement angle 2θ was 5–50°. FT-IR was performed on a Nicolet IS-10 (Thermo Fisher scientific, Waltham, MA, USA) with a scanning range of 4000–500 cm^−1^. The morphology of GO and the dispersion state of the fillers was observed using a Hitachi S-4800 scanning electron microscope.

The tensile property was tested on the CTM testing machine from Xieqiang Instrument Manufacturing (Shanghai, China) Co., Ltd. The tensile test standard was GB/T 528-2009, and the test spline was dumbbell shaped. The hardness was investigated according to GB/T 531.1-2008. The immersion tests can be seen in the Appendix A. Hot air accelerated aging tests were performed at 100 °C for 72 h in a rubber aging test box (MZ-401A), according to GB/T3512-2014. The results given represent an average value of at least five tests. Thermal analysis was tested in a thermogravimetric analyzer (TGA/DSC^3+^, Mettler-Toledo) under a nitrogen atmosphere, 50–600 °C, 10 K/min. The contact angle was investigated using a HARKE-SPCA contact angle measuring instrument.

The crosslinking density of the composites was measured through the swelling method in toluene, according to the Flory–Rehner equation (Equation (1)) [31], which is as follows: take a sample of about 1 g, weigh and record the mass M_0_, immerse it in a toluene solution and swell for 72 h in the dark, dry the reagent with filter paper, weigh and record the mass M_1_, then dry the sample after swelling at 80 °C to constant weight, and weigh and record the mass M_2_.
(1)$$Ve=-\frac{1}{Vs}\frac{\left[\ln(1-Vr)+Vr+\chi Vr^{2}\right]}{\left(Vr^{1/3}-Vr/2\right)}$$
in which V_e_ is the cross-link density of vulcanizates for a phantom network (mol/cm^3^), χ is the Flory–Huggins interaction parameter between rubber and solvent (0.43 for EPDM- toluene), Vs is the molar volume of toluene (106.5 cm^3^/mol for toluene), and V_r_ is the rubber volume fraction of the swollen sample at equilibrium swelling (the total volume of the sample does not include filler volume), which can be obtained through the following equation (Equation (2)):(2)$$Vr=\frac{M_{0}/\rho_{R}}{M_{0}/\rho_{R}+(M_{1}-M_{2})/\rho_{s}}$$
where *ρ****_S_*** is the density of toluene, *ρ****_R_*** is the density of unfilled EPDM, and *M**_R_*** is the mass of EPDM.

## 3. Results and Discussion

### 3.1. Characterization of GO, AGO, and HGO

The structures of GO, AGO, and HGO were tested with FT-IR and the spectrogram is displayed in Figure 2a. In comparison to GO, new peaks in AGO spectrum appeared at 1089 cm^−1^ and 1045 cm^−1^, following the stretching vibration of Si-O-C and Si-O-Si, respectively, indicating APTES has been covalently grafted onto the surface of GO sheets. In the spectrum of AGO, new peaks appeared at 2918 cm^−1^ and 2846 cm^−1^, which were -CH_2_-asymmetric and symmetric stretching vibration peaks, respectively. The characteristic peak at 1642 cm^−1^ indicated the formation of an amide (NHCO) bond between GO and ODA, and the peak at 1567 cm^−1^ further indicated the formation of an NHCO bond. The disappearance of the C-O-C stretching vibration peak at 1056 cm^−1^ indicated that the nucleophilic substitution reaction occurred between ODA and epoxy group on GO, providing that ODA has been grafted onto the GO sheet.

Through the XRD spectrum, the interlayer spacing and crystal structure can be obtained. Figure 2b is the XRD pattern of GO, AGO, and HGO. The XRD curve of GO showed a strong (002) characteristic diffraction peak at 10.1°, the spacing between GO sheets was calculated to be 0.83 nm using Bragg’s law. As for AGO, the intensity of the (002) characteristic diffraction peak decreased to a certain extent and shifted to 9.18°, and the interlayer spacing corresponding to this angle was 0.92 nm, indicating that APTES molecular chain was adsorbed on the surface of GO, which increased the interlayer spacing of AGO. After ODA modification, a weak peak of HGO appeared at 2θ = 5.02°, the peak intensity decreased, and the interlayer spacing increased from 0.83 nm to 1.76 nm. The increased interlayer spacing indicated that the long chain of ODA was covalently grafted into GO, and a reaction occurred between GO and ODA.

Figure 2c is the Raman spectrum of GO, AGO, and HGO. The Raman spectra has obvious characteristic peaks at 1353 cm^−1^ and 1597 cm^−1^, respectively, which correspond to D peak and G peak. It is generally believed that the G peak is generally derived from the E_2_g mode of sp^2^ carbon and the D peak corresponds to structural defects and dislocations. The intensity ratio of D peak to G peak (I_D/_I_G_) is generally considered to be an effective index to judge the structural disorder of carbon materials. The I_D/_I_G_ value of GO was 0.95. After APTES and ODA modification, the I_D/_I_G_ value of AGO and HGO increased to 1.13 and 1.01, respectively, indicating that the modification destroyed the structure of GO and made the structure of GO more disordered.

In addition, the morphology of GO, AGO and HGO were characterized by SEM, and the results are shown in Figure 3. Figure 3d shows that GO is wrinkled under the scanning electron microscope, Figure 3e shows that AGO has obvious wrinkled morphology, and the lamellae of HGO are larger and stacked together in Figure 3f. In summary, functionalized AGO and HGO were successfully synthesized by APTES and ODA modified GO.

### 3.2. Dispersion of the Fillers

The dispersion of the fillers in EPDM insulations was further observed by SEM. It is of great significance that the fillers uniformly disperse in the matrix. As shown in Figure 4b, GO is unevenly dispersed in EPDM matrix and a small number of aggregates appeared. We can see that neither agglomeration existed in Figure 4c or Figure 4d, indicating both AGO and HGO are well dispersed in EPDM and that they have good interfacial interaction and good compatibility with EPDM. Uniformly dispersed AGO and HGO can delay the diffusion path of plasticizers and increase its diffusion difficulty, while improving the mechanical properties of the insulations.

### 3.3. Anti-Migration Performance

The concentration of mixed plasticizers migrating to EPDM insulations was assessed using the immersion method. The immersion method involves the soaking of the material test piece in the liquid of the absorbed component, removing the test piece regularly for weighing, and calculating the weight gain rate and diffusion coefficient (the limit of migration) at the swelling equilibrium, which reflects the anti-migration performance. NG is unstable at room temperature and tends to explode, so we replaced it with triacetin and ethyl nitrate, which are similar in structure. The mixed plasticizers contains 30% DOS, 30% DBP, 20% triacetin, and 20% ethyl nitrate. The migration concentration curves of mixed plasticizer in EPDM insulations are shown in Figure 5, and Table 1 lists the migration equilibrium concentration. As shown in Figure 5a, the migration equilibrium concentrations of mixed plasticizer in the pure EPDM insulations at 25 °C was only 24.69%. At the same temperature, the migration equilibrium concentrations of mixed plasticizer in the GO/EPDM insulations decreased to 21.26% due to the barrier properties of GO. The equilibrium concentrations continues to decrease with the addition of AGO and HGO, the addition of functionalized GO are useful to increase the difficulty of mixed plasticizer diffusion, which significantly improved the migration equilibrium concentration. The equilibrium concentrations of AGO/EPDM insulation at 25 °C were 20.28%, indicating that the introduction of AGO can enhance the anti-migration performance. AGO/EPDM showed the best migration resistance caused by the silane coupling agent of APTES enhanced the compatibility between AGO and EPDM. In addition, after APTES modification, AGO dispersed more uniformly in EPDM matrix. The migration resistance of HGO/EPDM also shows a certain increase due to the introduction of HGO. This is due to the decrease in polar sites of HGO after modification, and the alkyl chain on the surface of HGO is easy to entangle with an EPDM molecular chain, which improves the interface conditions and blocks the diffusion of plasticizer. Moreover, the increased crosslink density of AGO/EPDM and HGO/EPDM also has an effect on reducing the amount of migration. With the increase in temperature, the equilibrium concentration of mixed plasticizers in the insulation increases, but the trend is consistent with that at 25 °C, and AGO/EPDM and HGO/EPDM maintain the excellent migration resistance of mixed plasticizers at a high temperature.

The equilibrium concentration represents the limit of plasticizer diffusion and migration, while the diffusion coefficient reflects the speed of plasticizer migration. For the sake of intuitive observation and comparation of the anti-migration performance in EPDM insulations, the diffusion coefficients were calculated according to Fick’s law [32]. Based on the diffusion coefficients listed in Table 2, as the temperature increased, the migration coefficient of various insulations increased due to the molecules reacting more violently at higher temperatures. The diffusion coefficient in proportion to the activities of plasticizers, the more active the plasticizer is, the higher the coefficient is, and the faster the diffusion and migration. At the same temperature, the plasticizer in the AGO/EPDM had the lowest diffusion coefficient. In addition, the diffusion coefficients of different samples at the same temperature are in the following order: EPDM, GO/EPDM, HGO/EPDM, and AGO/EPDM. This is consistent with the trend of equilibrium concentration. The diffusion coefficients of AGO/EPDM and HGO/EPDM are less than that of GO/EPDM, and the anti-migration performance of AGO/EPDM and HGO/EPDM both improved, proving that the modification of GO is conducive to preventing the diffusion of plasticizers.

In addition to the equilibrium concentration and diffusion coefficient, the migration activation energy can directly reflect the difficulty of plasticizer migration, the smaller the migration activation energy, the easier the plasticizer migrates, and the larger the diffusion coefficient. As shown in Figure 6, we obtained the migration activation energy according to Arrhenius equation [31]. The plot of lnD against 1000/RT shows a good linear relationship, and the correlation coefficients are all greater than 96%. We can see that the migration activation energy of AGO/EPDM was the largest in Table 3, which means its anti-migration performance was the best. The migration activation energy of HGO/EPDM is also larger than GO/EPDM, which proves that AGO and HGO again enhanced the anti-migration performance.

### 3.4. Thermal Analysis

The DOS content in the EPDM insulation after immersion can also reflect the anti-migration performance, the DOS content in different EPDM insulations was tested by thermal analysis. Figure 7 shows the TG and DTG curves of the EPDM insulation after immersion. In the TG curves of Figure 7a, we can see that DOS decomposed from 180 °C to 340 °C. Pure EPDM has a one-step decomposition from 180 °C to 480 °C, while EPDM insulation decomposed in two stages after immersion: 200–340 °C and 340–480 °C. This is due to the migrated DOS in EPDM insulation. The DTG curves of Figure 7b also confirmed this. We know that EPDM did not react with DOS, so we were able to obtain the migrated DOS content in EPDM insulations by subtracting the mass loss of pure EPDM from the total mass loss in the first stage. The DOS content in the EPDM and GO/EPDM insulation were 27.8% and 20.6%, respectively. After the introduction of functional GO, the DOS migration was further decreased, the DOS content in AGO/EPDM and HGO/EPDM were 17.5% and 18.4%. The thermal analysis was consistent with the immersion test results, indicating that the introduction of functional GO was beneficial to again preventing the migration of plasticizers.

### 3.5. Contact Angle Characterization

The affinity between the surface of the EPDM insulation and the plasticizer was directly tested by the contact angle to further indicate that the introduction of functional GO was beneficial to prevent the migration of plasticizer. To reduce the measurement error, when measuring the contact angle of the samples, three different points were taken, and the final result was averaged. Figure 8 was the contact angle of the insulation. The test liquid was DI and DOS. It can be seen from Figure 8a that the water contact angles of EPDM and GO/EPDM were 87° and 90°, respectively, and the water contact angle change is not obvious. The water contact angles of AGO/EPDM and HGO/EPDM increased to 99° and 96°, respectively, and hydrophobicity was improved. DOS was used as a test liquid to explore the affinity between the EPDM insulation surface and DOS. The DOS contact angle of EPDM was 21°, and the DOS contact angles of GOEPDM, AGO/EPDM, and HGO/EPDM were increased to 29°, 42°, and 35°, respectively. The reason for the improvement of AGO/EPDM performance is that the amino group is introduced on the surface, which reduced affinity with DOS, so it is also beneficial to preventing plasticizer diffusion. The introduction of HGO causes the surface of GO to create long, fat chains, which is also beneficial for decreasing affinity with DOS and preventing plasticizer migration.

### 3.6. Mechanical Properties

The insulation also requires good mechanical properties to protect SRM. Figure 9a demonstrates the stress–strain curves, and the tensile properties are shown in Figure 9b. Tensile stress–strain is a widely used test, and the tensile properties of EPDM, including tensile strength and elongation at break, can be obtained by recording the changes between stress and strain during stretching. The tensile strength of pure EPDM was 5.81 MPa. The tensile strength of AGO/EPDM and HGO/EPDM increased to 8.93 MPa and 7.96 MPa, respectively. Obviously, the addition of GO, AGO, and HGO all enhanced the tensile strength. Compared with GO/EPDM, the tensile properties of AGO/EPDM and HGO/EPDM are significantly improved.

The crosslink density and hardness of the insulation are shown in Figure 9c. Crosslink density is an important characteristic of polymers, exerting an essential influence on the comprehensive performance. It is obvious that the change trend of tensile properties is consistent with that of crosslinking density. Figure 9c exhibits the hardness of the insulations. It can be seen that the introduction of GO, AGO, and HGO improve the hardness. The maximum value of hardness of AGO/EPDM can reach 71°, while the value of EPDM is only 54°. The introduced AGO is uniformly dispersed in the matrix, which is the main reason that the hardness of the insulation materials is enhanced; meanwhile, the silane coupling APTES improves the compatibility between AGO and EPDM. As for HGO/EPDM insulation, the hardness is also improved, and the long alkane chain of ODA has better compatibility with the polymer and improves the compatibility of HGO with EPDM.

During the process of manufacturing, storage, transporting, and application, the insulation is prone to aging and degradation due to photo, thermal, and oxygen actions, so it is necessary to test the aging resistance of the composites. Tensile strength retention rate (K) was used to demonstrate the aging resistance, which was ratio of tensile properties of samples before and after aging, and a large K value means better aging resistance. The mechanical performance retention rates used to measure aging resistance were repeated three times for each group of samples. In Figure 9d, AGO/EPDM has the largest K value, followed by HGO/EPDM. Both the K values of AGO/EPDM and HGO/EPDM were higher than those of EPDM and GO/EPDM, which obviously enhanced the aging resistance. AGO and HGO make the diffusion path of oxygen and reactive radicals in the matrix more tortuous, thus enhancing the aging-resistant property.

As shown in Figure 10, the polygon area can be used to evaluate the comprehensive properties of different insulations, including equilibrium concentration, tensile properties, aging resistance, hardness, and crosslink density. The comprehensive performance of EPDM insulation is the worst, and the corresponding polygon area is the smallest. After the addition of functionalized GO, the performance is greatly improved, and the area of the polygon becomes larger. AGO/EPDM has the largest area, the best comprehensive performance, and excellent tensile strength, crosslink density, hardness, and aging resistance. The results show that functionalized GO is a candidate material for the enhancement of the comprehensive performance of EPDM insulation.

## 4. Conclusions

With the aim of enhancing the anti-migration performance and mechanical properties of EPDM insulation through functionalized GO, four different insulations were fabricated. Compared with pure EPDM at 25 °C, the migration equilibrium concentrations of AGO/EPDM and HGO/EPDM samples decreased by 33.26% and 28.16%, respectively. The immersion tests results revealed that functionalized GO is better for anti-migration performance, and AGO showed the best migration resistance. The results of mechanical properties show that the tensile properties, crosslink density, hardness, and aging resistance all increased with the introduction of AGO and HGO. The research presented here proves that it is useful to enhance the anti-migration performance and mechanical properties of EPDM insulation through functionalized GO in the SRM.

## Figures and Tables

**Figure 1 polymers-15-01731-f001:**
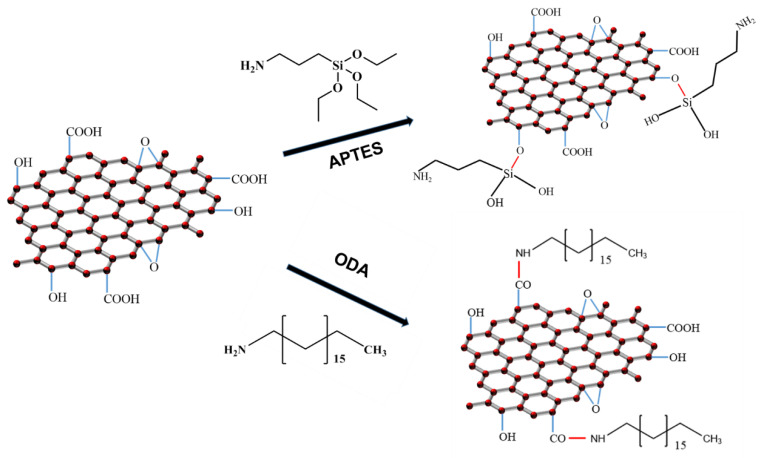
Synthesis of AGO and HGO.

**Figure 2 polymers-15-01731-f002:**
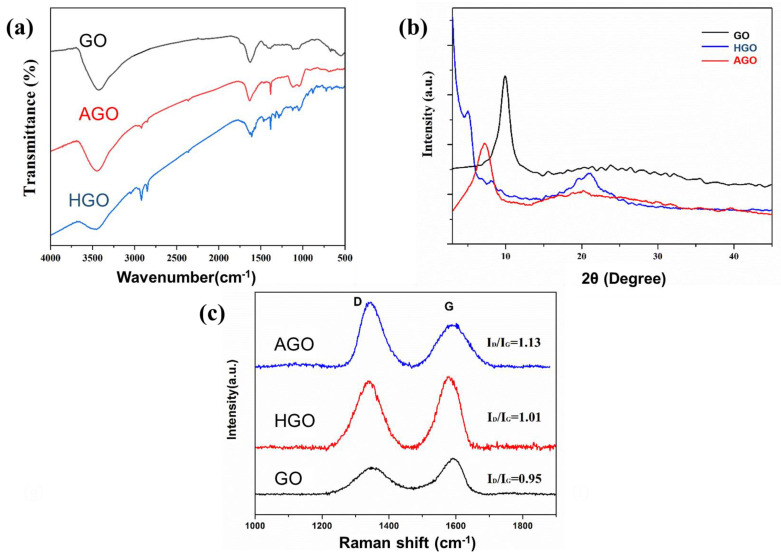
FTIR spectrum (**a**), Raman spectrum (**b**), and XRD patterns (**c**) of GO, AGO, and HGO.

**Figure 3 polymers-15-01731-f003:**
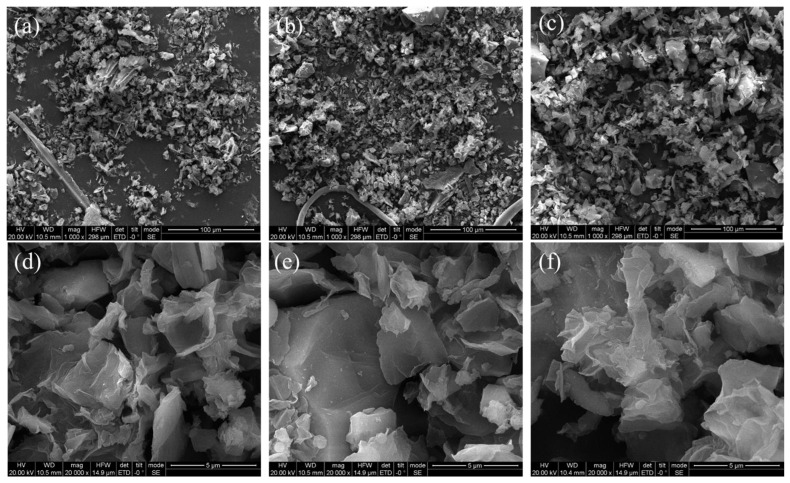
SEM images of GO (**a**,**d**), AGO (**b**,**e**), and HGO (**c**,**f**).

**Figure 4 polymers-15-01731-f004:**
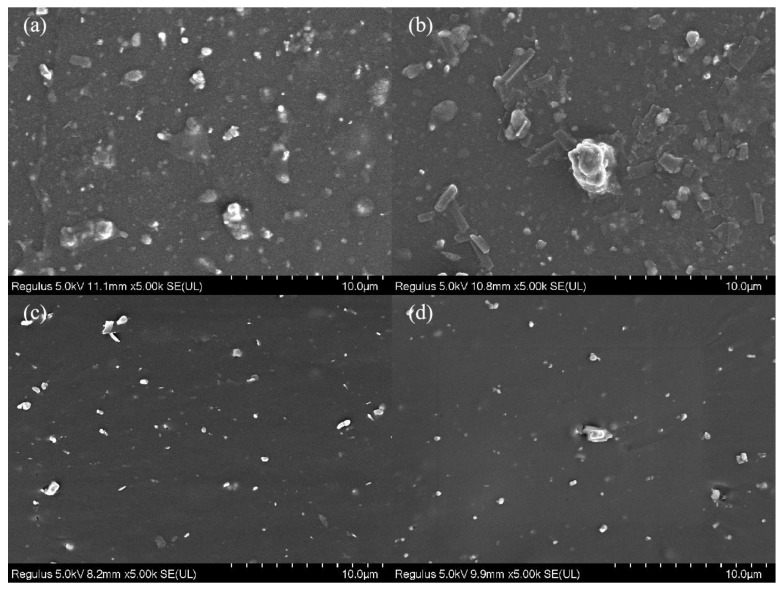
SEM images of (**a**) EPDM, (**b**) GO/EPDM, (**c**) AGO/EPDM, and (**d**) HGO/EPDM.

**Figure 5 polymers-15-01731-f005:**
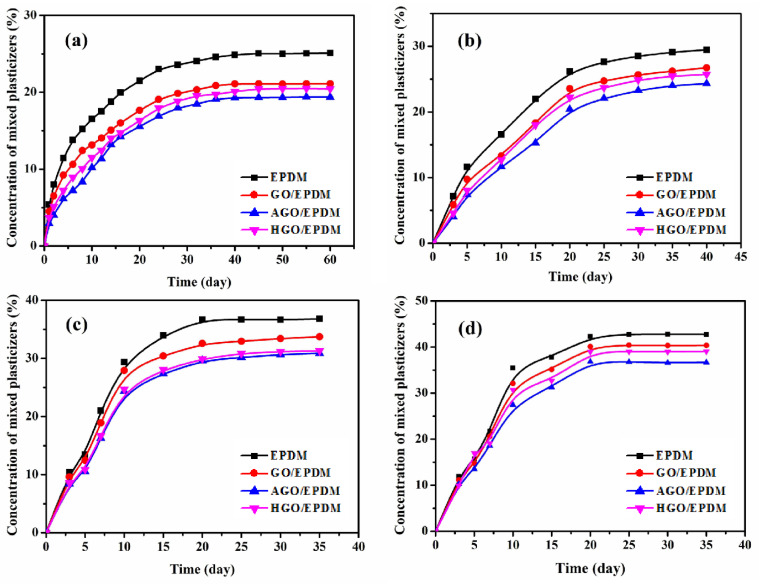
Time-varying concentrations of mixed plasticizers at different temperatures in the insulation: (**a**) 25 °C, (**b**) 40 °C, (**c**) 60 °C, and (**d**) 80 °C.

**Figure 6 polymers-15-01731-f006:**
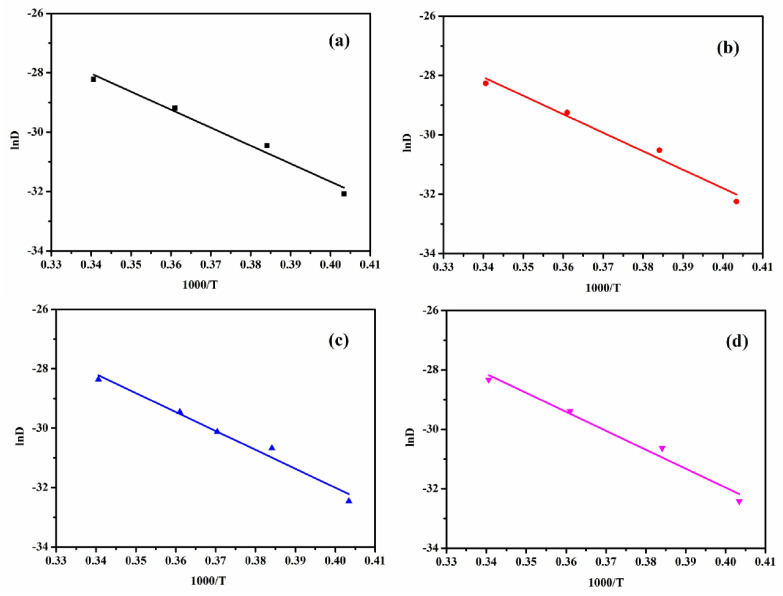
Linear fit of the activation energy in different insulation. (**a**) EPDM, (**b**) GO/EPDM, (**c**) HGO/EPDM, and (**d**) HGO/EPDM.

**Figure 7 polymers-15-01731-f007:**
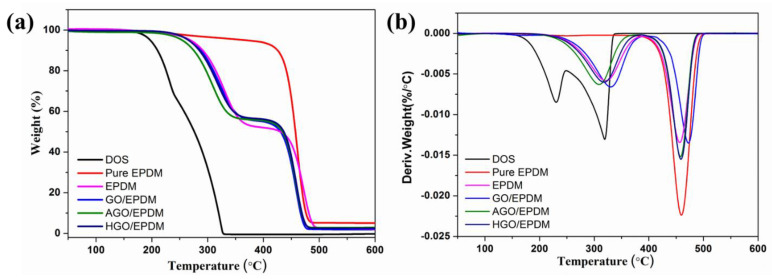
Thermal analysis of the EPDM insulation, TG (**a**) and DTG (**b**) curves.

**Figure 8 polymers-15-01731-f008:**
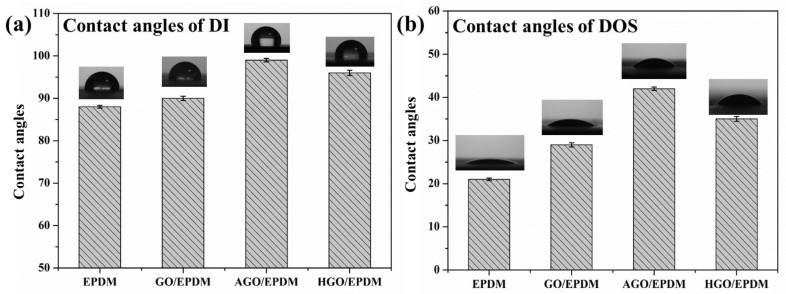
Contact angles of DI (**a**) and DOS (**b**).

**Figure 9 polymers-15-01731-f009:**
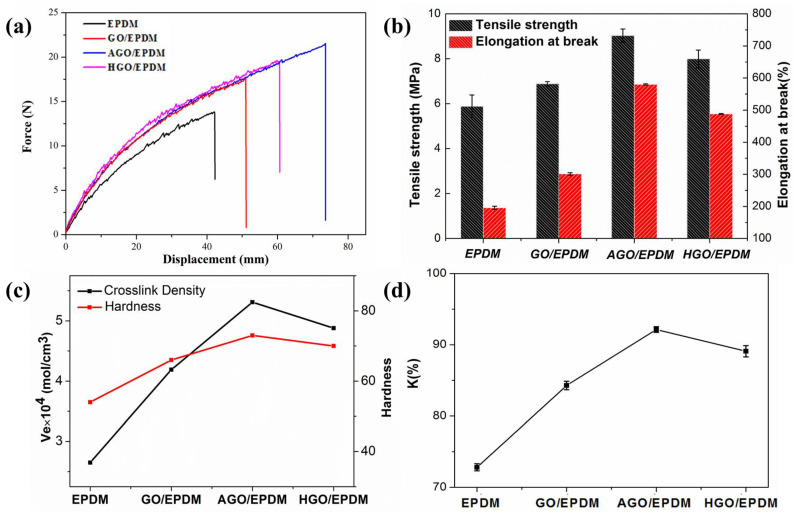
(**a**) The strain–stress curves, (**b**) tensile properties, (**c**) crosslink density and hardness, and (**d**) the mechanical performance retention rate of the EPDM insulations.

**Figure 10 polymers-15-01731-f010:**
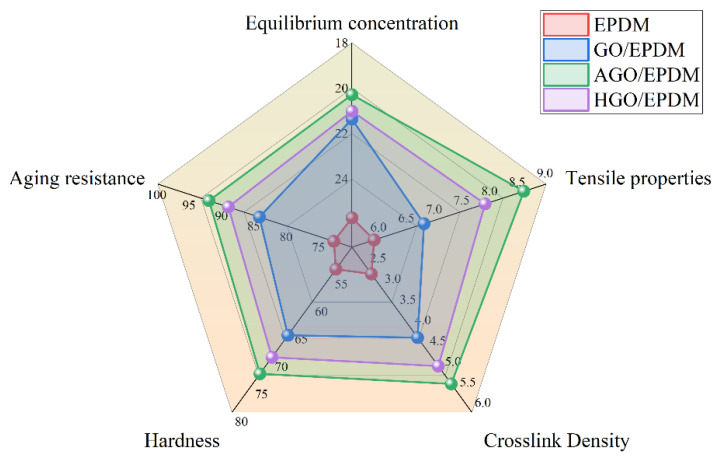
Performance polygon graph.

**Table 1 polymers-15-01731-t001:** Equilibrium concentration of the mixed plasticizers in the insulation.

Samples	25 °C/%	40 °C/%	60 °C/%	80 °C/%
EPDM	25.71	28.86	36.48	45.11
GO/EPDM	21.36	26.68	33.25	41.82
AGO/EPDM	20.28	24.35	31.17	40.93
HGO/EPDM	21.06	25.91	31.48	41.26

**Table 2 polymers-15-01731-t002:** Summary of diffusion coefficients (m^2^/s).

Samples	25 °C	40 °C	60 °C	80 °C
EPDM	4.10 × 10^−14^	1.73 × 10^−13^	7.12 × 10^−13^	1.54 × 10^−12^
GO/EPDM	2.92 × 10^−14^	1.47 × 10^−13^	5.95 × 10^−13^	1.49 × 10^−12^
AGO/EPDM	2.47 × 10^−14^	1.18 × 10^−13^	5.16 × 10^−13^	1.31 × 10^−12^
HGO/EPDM	2.60 × 10^−14^	1.43 × 10^−13^	5.90 × 10^−13^	1.42 × 10^−12^

**Table 3 polymers-15-01731-t003:** Migration activation energy in different insulation.

Samples	E_a_ (kJ/mol)	r^2^
EPDM	58.07	0.97
GO/EPDM	61.47	0.96
AGO/EPDM	63.46	0.98
HGO/EPDM	62.17	0.96

## Data Availability

Data will be made available on reasonable request.

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
