# Peer review of "Enhancing the Anti-Migration Performance and Mechanical Properties of EPDM Insulation through Functionalized GO"

_polymers, 2023, doi:10.3390/polym15071731_

Round 1

Reviewer 1 Report

The anti-migration properties and migration kinetics of EPDM insulations filled with GO, 3-Aminopropyltriethoxysilane 18 modified GO (AGO), and octadecylamine modified GO (HGO) were studied. In addition, the mechanical properties, including the tensile properties, crosslink density, hardness, and aging resistance of different EPDM insulations, were explored.

 Chapter 2.1: Is there any reason to put the material information in the supporting file, not the paper text? I can understand this in the case of the preparation of AGO and HGO. Still, I would expect a basic specification of the material more directly in the text of the article.

 Row 112: It is written: “The crosslinking density of the composites was measured by swelling method, the specific 111 process is shown in Supporting Information[29]“ Not only in supporting material but also in chapter 3.3. Consider including a description directly in chapter 2.4.

 Row 119: Is it really GA? I can see only AGO and HGO spectra with these spectral bands (2918 cm-1 and 2846 cm-1) in Figure 2a.

 Row 222: Comment to Fig. 6 is mentioned, but Fig. 6 is missing in the paper. Probably Fig. 7 should be this one.

 Fig. 8b: I suppose the black column is Tensile strength and the red one elongation. It would be helpful to add a legend similar to Fig. 8c.

The error bars are also visible on the graph (Fig. 8b). Is it possible to specify how many samples were used for the test for each material (frequency of measurements)? Similarly, concerning Fig. 8d. Suppose the rest results are with the count of one? Please specify all methods.

Row 258 – 267: Aging resistance. Is it possible to describe the aging process (increased temperature, photodegradation, or oxidation)? Could you specify levels and times in detail, or is it only based on the ability of AGO and HGO to make the diffusion path of oxygen and reactive radicals in the matrix more tortuous?

Reviewer 2 Report

The authors studied enhancing anti-migration performance and mechanical properties of the EPDM insulation material using functionalized graphene-oxide (AGO/HGO). They provided characterization of the synthesized composite material and performed anti-migration performance evaluations using immersion method. They found the addition of GO and functional GO can improve the anti-migration performance of EPDM, not only that, the addition of the extra components significantly improved the mechanical performance of the insulation material. By detailed characterization of the microscopic structure of the composite material, they found that the functionalized GO show a better performance which is contributing to the dispersion of functionalized GO molecules in the EPDM substrate. This work provide a systematic study on how to enhance the anti-migration performance of EPDM materials and could provide a practical meaningfulness on similar materials. I would not like to recommend this paper to be published on Polymers in current form before the authors addressing the following issues:

(1)    The introduction of this study is weak. The authors should review enough recent studies on similar systems on how to improve the anti-migration performance of insulation material via modification of extra components. Please provide rational discussions on how other studies mentioned the anti-migration mechanism on molecular level.

(2)    The anti-migration performance from the immersion method results (fig 5) do show an improvement of the equilibrium concentration of mixed plasticizers but not significant. I was expecting the addition of the GO or functionalized GO could lead to a dramatic decrease in the small molecule swelling but the improvement seems only around 10%. Does this improvement come from the addition of the extra components, which occupied extra volume? If it is true then, we could basically adding anything to the EPDM and improve the anti-migration performance via immersion method. I highly recommend authors provide other proof that showing GO/functionalized GO do have this anti-migration property instead of just increasing the total weight and make it mathematically looks like own some anti-migration properties.

(3)    The addition of GO/functionalized GO will definitely improved mechanical property due to the formation of chemical/physical crosslinkers. So this is pretty natural property that people could imagine when GO/functionalized GO were added to the system. I think this characterization is weak and boring. Therefore, please provide some other characterizations on the composite material to show such modification can really improve the properties of the EPDM insulation materials. For example: the influence of GO/functional GO modifications on the insulation properties. I think people will be interested in that instead of how GO improved mechanical properties, because we already knew it without your characterizations. Please provide something new.

Reviewer 3 Report

The article “Enhancing the Anti-migration performance of EPDM insulation through functionalized GO” by Zhehong Lu et al. is devoted to the synthesis and functionalization of graphene oxide and obtaining of new composites based on them to enhance the anti-migration performance of EPDM insulation.

The work is interesting and has scientific novelty. Article recommended for publication in Polymers Journal.

As the main remarks, it should be said that there are a large number of grammatical flaws in the work. In SI, the character t looks like an extra letter in the text, so italicize the character or put it in brackets. In the SI text, it is necessary to correct superscripts, extra spaces between characters.

It is also not clear from the text why a four-necked flask was used in the experiment.

What is KH550? This is completely incomprehensible in SI.

Abstract should be corrected in the manuscript, namely “Herein, two functionalized graphene oxide (GO) was used to enhance the anti-migration performance of EPDM insulation, GO, 3-Aminopropyltriethoxysilane modified GO (AGO) and octadecylamine modified GO (HGO) were filled into EPDM to fabricate EPDM insulation".

In addition, a lot of grammatical errors (spaces, missing letters, etc.) were found in the text. The manuscript should be READ!

Round 2

Reviewer 2 Report

Publish as is.